# Effect of Commercial Microbial Preparations Containing *Paenibacillus azotofixans*, *Bacillus megaterium* and *Bacillus subtilis* on the Yield and Photosynthesis of Winter Wheat and the Nitrogen and Phosphorus Content in the Soil

**Arkadiusz Stępień** [1,*] **, Katarzyna Wojtkowiak** [2] **and Ewelina Kolankowska** [2]

1 Department of Agroecosystems and Horticulture, University of Warmia and Mazury in Olsztyn, pl. Łódzki 3, 10-719 Olsztyn, Poland

2 Department of Heavy-Duty Machines and Research Methodology, University of Warmia and Mazury in Olsztyn, ul. Oczapowskiego 11, 10-719 Olsztyn, Poland

* Correspondence: arkadiusz.stepien@uwm.edu.pl; Tel.: +48-895233266

**Abstract:** The present state of knowledge and biotechnological advances have allowed the potential of microorganisms to be used effectively in crop cultivation. A field study on the use of commercial bacterial preparations in the cultivation of winter wheat (*Triticum aestivum* L.) was carried out in the years 2017–2019 at the Educational and Experimental Station in Tomaszkowo (53°71′ N, 20°43′ E), Poland. This study analysed the effect of commercial microbial preparations containing *Paenibacillus azotofixans*, *Bacillus megaterium* and *Bacillus subtilis*, applied during the winter wheat growing season, on the grain yield, protein content, leaf greenness index (SPAD), the course of photosynthesis and the N-NO$_3$, N-NH$_4$ and P contents in the soil. The highest grain yield was noted following the application of mineral fertilisation and the three microbial preparations in combination (*Paenibacillus azotofixans*, *Bacillus megaterium* and *Bacillus subtilis*), as well as NPK with *Paenibacillus azotofixans*, in relation to mineral fertilisation alone (by 19.6% and 18.4%, respectively). The microbial preparations had a significant effect on the leaf greenness index (SPAD) at both test dates. No interaction was recorded between the years of study and the preparations applied on the SPAD values. The highest leaf photosynthetic index at both observation dates was noted for the application of NPK + *P. azotofixans*, as well as for NPK and all the preparations combined (*P. azotofixans*, *B. megaterium*, *B. subtilis*). The highest N-NO$_3$, N-NH$_4$ and P contents in the soil were obtained using NPK and all microbial preparations combined. Strong correlations were found between the SPAD index and the photosynthetic index value and the protein content in wheat grains and between the N-NO$_3$, N-NH$_4$ and P contents in the soil and the wheat grain yield.

**Keywords:** winter wheat; leaf greenness index (SPAD); protein; photosynthesis; nitrogen; phosphorus

## 1. Introduction

The design of stable winter wheat yields is determined by the use of proven cultivation methods and technologies under specific soil and climate conditions [1–3]. Intensive farming systems contribute to a reduction in soil fertility, which can result in environmental degradation as well as a drop in the quantity and quality of crop yields [4–6]. An alternative to conventional production resources used to support crop production may be biological preparations serving as biofertilisers and biopesticides. These are intended to protect plants against pathogens, influence soil fertility and shape the growth and development of plants [7–11]. An important group of biological preparations are those containing plant-growth-promoting rhizobacteria (PGPR), which comprise bacterial strains of *Azoarcus*, *Azospirillum*, *Azotobacter*, *Arthrobacter*, *Bacillus*, *Clostridium*, *Enterobacter*, *Gluconacetobacter*, *Pseudomonas* and *Serratia* [12]. The rhizosphere bacteria are supportive of the recovery of nutrients from the soil and are important for soil fertility [9,13–16].

Bacteria of the genus *Bacillus* are widely distributed in nature. Most of them are species that are safe for humans, animals and plants. Due to the above-mentioned characteristics of the bacteria *Bacillus* sp., they are used for the production of commercial preparations in the form of insecticides or biostimulants and are involved in supporting plant production [17–20]. *Bacillus* sp. *bacteria* (*B. subtilis*, *B. cereus*, *B. thuringiensis*, *B. pumilus*, *B. megaterium*, etc.) have evolved mechanisms to stimulate plant growth by increasing the availability of the nutrients: N, P, potassium (K) and iron (Fe). Moreover, *Bacillus* strains are capable of fixing molecular nitrogen [19,21–23].

*Bacillus subtilis* is a saprophyte which decomposes organic compounds of plant origin. This bacterium produces peptide antibiotics, e.g., polymyxin B and subtilin, amino acids, polysaccharide inulin and enzymes, e.g., amylase and protease [24–28]. It also forms siderophores (bacillobactin) which are capable of binding iron ions by binding all available forms of iron into chelates and sharing them with plants. The protein of these bacteria also contains hydrophobin BsIA, which reduces surface tension, thus increasing the wetting of the surface on which the bacteria are located, leading to an improvement in moisture within the root system and covering it with an additional protective film (which is particularly important during dry periods) [29–31]. It also stabilises soil colloids [32]. The bacterium *Bacillus subtilis* reduces the number of pathogenic fungi and bacteria in the soil, thus contributing to improved phytosanitary status in crops. [33,34]. *Bacillus subtilis* can also solubilise P in the soil, contribute to nitrogen fixation and produce siderophores which inhibit the growth of pathogens. *Bacillus subtilis* increases stress tolerance in its plant hosts by inducing the expression of stress response genes, phytohormones and stress-related metabolites [20,35]. *Bacillus megaterium* bacteria are described as soil microorganisms with a natural ability to produce acids or enzymes as metabolites, making them capable of dissolving phosphorus [36,37]. *Paenibacillus azotofixans* was initially classified as *Bacillus azotofixans* and then reclassified as belonging to the genus *Paenibacillus* [38]. *Paenibacillus* presents the nifH gene encoding the Fe nitrogenase protein, an enzyme responsible for nitrogen fixation [39]; it has been suggested as a potential biofertiliser for certain crops, e.g., maize or wheat [40–43]. What is more, *Paenibacillus* are well known as effective plant growth promoters in many crops, e.g., maize, wheat, or sorghum [39,44]. *Paenibacillus azotofixans* is a nitrogen-fixing bacterium often found in the soil and rhizospheres of various plants [45–49]. The use of biological preparations in the form of biofertilisers may be a way to ensure environmental stability in intensive agricultural production. To date, no studies have focused on the comprehensive effect of biological preparations on the photosynthesis processes, the yield of wheat grain and the content of nitrogen and phosphorus in soil.

The aim of the study was to assess the effect of commercial microbial preparations containing *Paenibacillus azotofixans*, *Bacillus megaterium* and *Bacillus subtilis*, applied during the winter wheat growing season, on the grain yield, protein content, leaf greenness index, the course of photosynthesis and the $N-NO_3$, $N-NH_4$ and P contents in the soil.

## 2. Materials and Methods

### 2.1. Experimental Design, Growing Conditions and Treatments

UPTOHERE The research experiment was carried out at the Educational and Experimental Station in Tomaszkowo (53°71′ N, 20°43′ E), Poland. The experiments were carried out in the years 2017–2019 using commercial bacterial preparations that are currently used in the cultivation of winter wheat (*Triticum aestivum* L.). The method employed was that of a static field experiment carried out in four replicates in a randomised block design. The area of the seeding plot was 9.90 m$^2$ and that of the harvest plot was 8.00 m$^2$. Winter wheat of the KWS OZON cultivar was sown annually in succession, with winter triticale as the nurse crop in 2016. The wheat was sown at a density of 500 seeds·m$^{-2}$, with a row spacing of 12.5 cm.

Wheat was cultivated on lessive soil with a granulometric composition of a medium dusty loam. The particle composition of the mineral surface soil horizon included strong loamy sand. The soil was slightly acidic (in KCl solution with pH 5.7), the carbon content

was 10.0 g kg$^{-1}$, and the total nitrogen content was 0.99 g kg$^{-1}$. The soil abundance in available nutrients was high for P (85.1 mg kg$^{-1}$) and medium for K (155.0 mg kg$^{-1}$). Weather conditions registered during field trials are presented in Table 1. The factor under research was the application of commercial bacterial preparations in combination with NPK mineral fertilisers (Table 2).

**Table 1.** Monthly air temperature and monthly rainfall in the 2016–2019 season. Meteorological data against the years 1981–2010 (Data obtained from the Meteorological Station at Tomaszkowo (53°71′ N, 20°43′ E), Poland).

| Growing Season | Mean Temperature (°C) | | | | | | | | |
|---|---|---|---|---|---|---|---|---|---|
| | IX * | X | XI-III | IV | V | VI | VII | VIII | Av. |
| 2016/2017 | 13.6 | 6.1 | 2.4 | 5.7 | 12.1 | 15.7 | 16.8 | 17.4 | 7.5 |
| 2017/2018 | 12.8 | 8.7 | 3.9 | 10.8 | 15.7 | 17.2 | 19.7 | 19.2 | 8.6 |
| 2018/2019 | 14.5 | 8.7 | 3.3 | 8.0 | 11.6 | 20.2 | 17.1 | 18.5 | 8.8 |
| 1981–2010 | 12.8 | 8.0 | 2.9 | 7.7 | 13.5 | 16.1 | 18.7 | 17.9 | 7.9 |
| | Rainfall (mm) | | | | | | | | Sum |
| 2016/2017 | 21.1 | 104.3 | 84.8 | 59.1 | 25.1 | 74.5 | 107.6 | 63.1 | 693.8 |
| 2017/2018 | 168.1 | 114.9 | 42.4 | 33.5 | 25.0 | 53.7 | 141.0 | 44.6 | 713.4 |
| 2018/2019 | 20.3 | 84.7 | 16.0 | 0.0 | 142.8 | 120.6 | 56.3 | 55.9 | 677.6 |
| 1981–2010 | 56.9 | 42.6 | 44.8 | 33.3 | 58.5 | 80.4 | 74.2 | 59.4 | 581.8 |

* Month/Phenological growth stages (BBCH scale): IX/germination–leaf development (BBCH 00–19); X/tillering (BBCH 20 . . . ); XI-III/winter dormancy; IV/starting vegetation; V/stem elongation–booting (BBCH 30–49); VI/Inflorescence emergence, heading–flowering, anthesis (BBCH 51–69); VII/development of fruit–senescence (BBCH 71–99); VIII/harvesting.

**Table 2.** Design of the field experiment. Dose and date of application microbial preparations used in the field experiment.

| Treatment (Shortcut) | Component | Application Date/Dose |
|---|---|---|
| NPK (NPK) | N (ammonium sulphate 34%; <br> N (ammonium sulphate 34%; <br> P (triple superphosphate 20%); <br> K (potash salt, 49.8%) | BBCH$_{23–24}$/90.0 kg ha$^{-1}$ <br> BBCH$_{31–32}$/60.0 kg ha$^{-1}$ <br> Pre-sowing/70.0 kg ha$^{-1}$ <br> Pre-sowing/100.0 kg ha$^{-1}$ |
| NPK * + *Paenibacillus azotofixans* (NPK + PA) | *Paenibacillus azotofixans* <br> $1 \times 10^9$ CFU ** in 1 g of the product (maltodextrin) | BBCH$_{23–24}$/1.0 L ha$^{-1}$ <br> BBCH$_{31–32}$/1.0 L ha$^{-1}$ |
| NPK * + *Bacillus megaterium* (NPK + BM) | *Bacillus megaterium* <br> $1 \times 10^9$ CFU in 1 g of the product (maltodextrin) | BBCH$_{23–24}$/1.0 L ha$^{-1}$ <br> BBCH$_{31–32}$/1.0 L ha$^{-1}$ |
| NPK * + *Bacillus subtilis* (NPK + BS) | *Bacillus subtilis* <br> $5 \times 10^9$ CFU in 1 g of the product (maltodextrin) | BBCH$_{23–24}$/1.0 L ha$^{-1}$ <br> BBCH$_{31–32}$/1.0 L ha$^{-1}$ |
| NPK * + *Paenibacillus azotofixans* + *Bacillus megaterium* + *Bacillus subtilis* (NPK + PA + BM + BS) | *Paenibacillus azotofixans* <br> *Bacillus megaterium* <br> *Bacillus subtilis* | BBCH$_{23–24}$/1.0 L ha$^{-1}$ BBCH$_{31–32}$/1.0 L ha$^{-1}$ <br> BBCH$_{23–24}$/1.0 L ha$^{-1}$ BBCH$_{31–32}$/1.0 L ha$^{-1}$ <br> BBCH$_{23–24}$/1.0 L ha$^{-1}$ BBCH$_{31–32}$/1.0 L ha$^{-1}$ |

* NPK—mineral fertilisers were applied on all plots at the same doses and dates as on the NPK plot (control), ** CFU—colony-forming unit.

The sowing, cultivation treatment and harvesting of the wheat were carried out in accordance with the agrotechnical requirements specific to the plant species. No protection against pests or diseases was performed. Weeds were controlled using herbicides: BBCH 31–32–Axial 50 EC 0.8 L/ha (pinoxaden)–50 g/L (5.05%), Mustang Forte 195 SE 1.0 L/ha (Florasulam–5 g/L Aminopyralid–10 g/L, 2,4-D–180 g/L).

### 2.2. Yield and Quality Analysis Samplings

The wheat grain was harvested during the first ten-day period of August using a plot harvester (Wintersteiger Classic 1540, Austria). The wheat grain yield was determined at a moisture of 15%. The protein content was determined on 1.0 kg of samples using a NIR System Infratec 1241 Analyzer camera (Foss, Hillerod, Denmark).

### 2.3. Leaf Gas Exchange, SPAD Index Measurement

During the winter wheat growing season, the net photosynthetic intensity [mmol $CO_2 \cdot m^{-2} \cdot s^{-1}$] and the leaf greenness index (SPAD) were assessed at two different times:

(1)    stem elongation–first node at least 1 cm above tillering node ($BBCH_{31}$)
(2)    inflorescence emergence, heading–beginning of heading ($BBCH_{51}$).

The measurements of photosynthesis were carried out on a fully developed, youngest leaf on five randomly selected plants. Ten records were made on each leaf at 5-s intervals. The measurement was performed on a sunny day, in the forenoon hours (09:00 a.m.–11.00 a.m.). Photosynthesis was assessed using an LCi compact camera (ADC BioScientific LCi Analyser Serial No. 32568) manufactured by Eijkelkamp.

The measurements of the leaf greenness index (SPAD) were taken on a fully developed, youngest leaf on ten randomly selected plants. The leaf greenness index was assessed using a SPAD–502 Plus camera by Konica Minolta.

### 2.4. Physicochemical Soil Analyses

Soil pH was measured potentiometrically in 1M KCl in soil suspension to a 1:5 solution. Total N was determined by mineralisation of a sample with sulphonic acid with an addition of a catalyst (Se mixture), the distillation of products of sodium hydroxide reaction, followed by titration with 0.1 M hydrochloric acid solution against the Tashiro indicator. $NH_4^+$-N and $NO_3^-$-N was determined calorimetrically, with Nessler's reagent and with phenyldisulphophenolic acid as colouring agents, respectively (UV-1201 V spectrophotometer, Shimadzu Corporation Kyoto, Japan). Available phosphorus (P) and potassium (K) in the soil (mg/kg) was measured by the Egner-Riehm method in an aqueous solution containing calcium lactate (($CH_3CHOHCOO)_2Ca$) acidified with hydrochloric acid to pH 3.6. Organic C content was determined by oxidation with potassium dichromate in sulphonic acid solution and measurement of the absorbance on a spectrophotometer UV-1201 V (Shimadzu Corporation, Kyoto, Japan).

### 2.5. Statistical Analysis

Results were statistically analysed using the Statistica 13.1 PL statistical software package. The differences between the experimental plots were determined using a one-way analysis of variance (ANOVA), and Tukey's test was applied to identify homogeneous groups. The calculations were made at a significance level $\alpha = 0.05$ [50]. Using a regression analysis, equations describing the photosynthetic process as determined by the temperature and rainfall occurring at the $BBCH_{31}$ and $BBCH_{51}$ stages were determined [51]. The correlation of the influence of the applied biological preparations on the leaf greenness index, photosynthesis, grain yield, protein content in grains and the N-$NO_3$, N-$NH_4$ and P contents in the soil in wheat cultivation was also determined. To this end, the principal component analysis (PCA) method was used to determine the strength and direction of the correlation between the measurement variables [52].

## 3. Results and Discussion

### 3.1. Grain Yield and Protein Content

The lowest yield obtained in 2018 results from the unfavourable rainfall distribution during this growing season (Figure 1). Excessive rainfall during the sowing and emergence of wheat in 2017 (September–October) might have contributed to the poor rooting of the plants, which could have resulted in lower yields in 2018. Furthermore, low rainfall during



winter (February–March 2018), as well as water shortages during the intensive growth (May–June 2018), adversely affected the growth and development of the plants. During this period, the unfavourable water balance was exacerbated by the occurrence of higher temperatures. A reduced yield is a typical response to water stress, as both the photosynthetic intensity and the plant growth processes are reduced [53–57]. High temperatures combined with low rainfall increase the loss of nutrients (N, P, K), particularly in poor soils [58]. Adequate weather conditions occurred in the growing seasons of 2016/2017 and 2018/2019, which resulted in higher yields (by 2.64 t and 1.72 t ha$^{-1}$, respectively) as compared to the season of 2017/2018. A higher yield can primarily result from a relatively normal temperature and the appropriate distribution of the optimum rainfall amount during the May-June period, which is critical for cereal growth in Poland [59]. Irrespective of the years, as compared to the control plot (NPK), the highest grain yield was noted following the application of the three combined microbial preparations (NPK + *P. azotofixans* + *B. megaterium* + *B. subtilis*), as well as NPK + *P. azotofixans* by 19.6% and 18.4%, respectively. In the years of the study, the effect of the application of the microbial preparations on the grain yield was not uniform (Table 3). In a study by Turan et al. [60], the introduction of *Bacillus subtilis*, *Bacillus megaterium* and *Azospirillum brasilense* into the soil increased the grain yield by 24%, 19% and 19%, respectively, while their application in combination resulted in an increase in the grain yield by 33% as compared to the control (with no microbial preparations). In the authors' studies in 2017 and 2019, the highest grain yields were obtained following the application of NPK + *P. azotofixans* and NPK and all the preparations combined (*P. azotofixans* + *B. megaterium* + *B. subtilis*), while in 2018, optimal yields were observed only following the application of all the three preparations in combination. The preparation with *P. azotofixans* was applied under optimal conditions for wheat growth and development, and all the preparations applied in combination performed well under extreme weather conditions thanks to their ability to supplement each other. In general, the effects of microbial preparations are more noticeable under unfavourable weather conditions [61] and less noticeable under optimal conditions prevailing during the crop-growing season [62]. Bacterial endophytes found in plants employ direct or indirect mechanisms to improve the growth and development of plants and increase their tolerance to biotic and abiotic stresses [63–65]. According to de Lima et al. [66], Ali and Khan [67], Maslennikova and Lastochkina [68], plants inoculated with *B. subtilis* can improve their growth under water stress conditions, mainly due to increased water use efficiency.

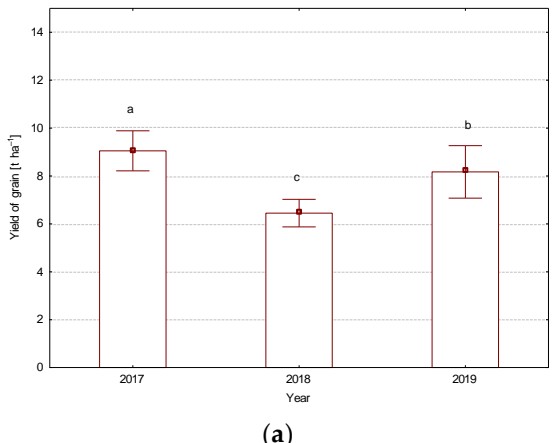

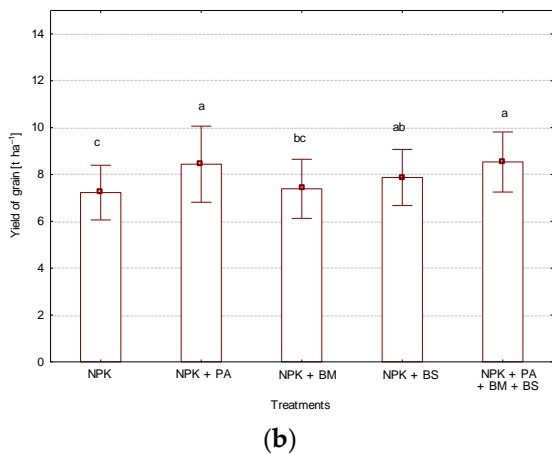

(**a**)　　　　　　　　　　　　　　　(**b**)

**Figure 1.** The winter wheat grain yield after application of microbial preparations (mean value and standard deviation), (**a**) average for the years, (**b**) average for treatment. Values followed by the same letters do not differ significantly in Tukey's (HSD) test ($p < 0.05$). PA—*Paenibacillus azotofixans*; BM—*Bacillus megaterium*; BS—*Bacillus subtilis*.

**Table 3.** The grain yield after application of microbial preparations (mean value and standard deviation), interaction between the years and treatments, t ha$^{-1}$.

| Year | NPK | NPK + PA * | NPK + BM ** | NPK + BS *** | NPK + PA + BM + BS |
|---|---|---|---|---|---|
| 2017 | 8.72 ± 0.56 ab | 9.81 ± 0.23 a | 8.38 ± 0.69 abc | 8.60 ± 0.58 abc | 9.76 ± 0.89 a |
| 2018 | 6.46 ± 0.32 e | 6.41 ± 0.84 e | 5.81 ± 0.29 e | 6.45 ± 0.17 e | 7.13 ± 0.07 cd |
| 2019 | 6.51 ± 0.38 de | 9.10 ± 0.60 ab | 7.98 ± 0.45 bcd | 8.57 ± 0.91 abc | 8.71 ± 0.75 ab |

* PA—*Paenibacillus azotofixans*; ** BM—*Bacillus megaterium*; *** BS—*Bacillus subtilis*. Values followed by the same letters do not differ significantly in Tukey's (HSD) test ($p < 0.05$).

The quality characteristics of wheat grains are determined by the genetic conditions and modified by the cultivation technology type and the weather conditions [1–3]. Certain microorganism species (plant growth-promoting rhizobacteria) can extract nutrients from the soil or the atmosphere and, consequently, contribute to improved plant nutrition [20,21,28,69]; however, these may not be sufficient quantities to satisfy all plant nutrient needs [70–72]. In the authors' own study, the highest protein contents in grains were noted in 2018; lower contents were observed in other years (Figure 2, Table 4). Jie et al. [73] showed that weather conditions, including the temperature, insolation duration and rainfall during the plant growth period, are the main factors that determine the wheat protein content and quality. Protein accumulation under high temperatures and no rainfall conditions is related to metabolism, transport, photosynthesis, responses to stress, detoxication and protein synthesis [74]. According to Yang et al. [75], Liu et al. [76] and Sehgal et al. [55], mild stress due to water shortage can promote the remobilisation of carbon assimilates into the grains, thus speeding up the grain filling and ultimately improving the yield quality. The influence of microbial preparations on the protein content in wheat grain was not demonstrated (Figure 2).

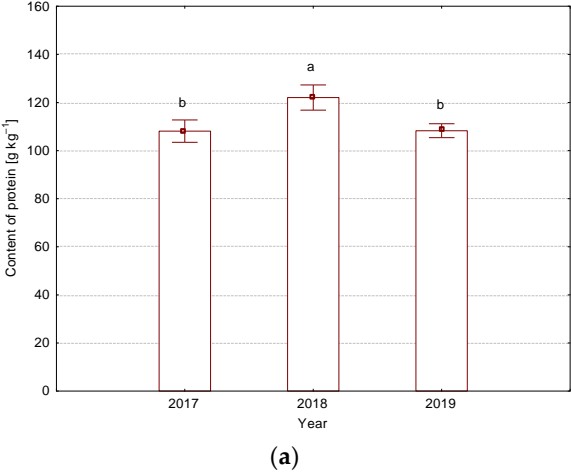

(**a**)

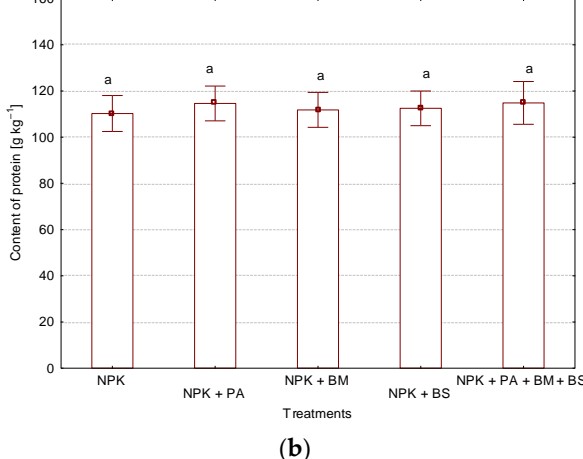

(**b**)

**Figure 2.** Protein content in wheat grains after application of microbial preparations (mean value and standard deviation), (**a**) average for the years, (**b**) average for treatment, g kg$^{-1}$. Values followed by the same letters do not differ significantly in Tukey's (HSD) test ($p < 0.05$). PA—*Paenibacillus azotofixans*; BM—*Bacillus megaterium*; BS—*Bacillus subtilis*.

**Table 4.** Protein content in wheat grains after application of microbial preparations (mean value and standard deviation), interaction between the years and treatments, g kg$^{-1}$.

| Year | NPK | NPK + PA * | NPK + BM ** | NPK + BS *** | NPK + PA + BM + BS |
|---|---|---|---|---|---|
| 2017 | 103 ± 2.36 [b] | 108 ± 4.43 [b] | 108 ± 2.44 [b] | 109 ± 5.57 [b] | 109 ± 3.30 [b] |
| 2018 | 121 ± 4.78 [a] | 121 ± 2.21 [a] | 121 ± 3.30 [a] | 121 ± 6.18 [a] | 124 ± 6.39 [a] |
| 2019 | 109 ± 2.50 [b] | 108 ± 3.56 [b] | 106 ± 3.77 [b] | 108 ± 1.50 [b] | 110 ± 2.89 [b] |

* PA—*Paenibacillus azotofixans*; ** BM—*Bacillus megaterium*; *** BS—*Bacillus subtilis*. Values followed by the same letters do not differ significantly in Tukey's (HSD) test ($p < 0.05$).

### 3.2. The Leaf Greenness Index (SPAD) and the Photosynthetic Rate

Weather conditions during the research had a significant effect on the SPAD index value only at the 1 node stage ($BBCH_{31}$), with its highest value noted in 2019 (Figure 3). The microbial preparations had a significant effect on the SPAD index value at both test dates ($BBCH_{31}$ and $BBCH_{51}$). Following the application of NPK and the preparation containing *P. azotofixans* and NPK with all the biopreparations (*P. azotofixans* + *B. megaterium* + *B. subtilis*), significantly higher values of the SPAD index were noted at both growing stages.

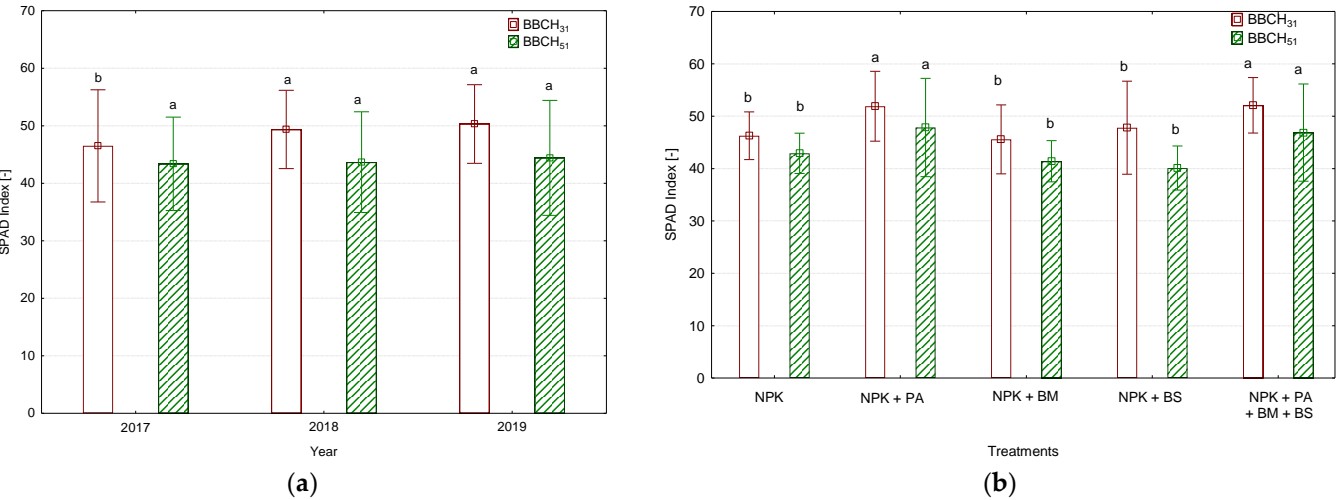

**(a)**                                        **(b)**

**Figure 3.** The leaf greenness index (SPAD) after application of microbial preparations (mean value and standard deviation), (**a**) average for the years, (**b**) average for treatment. Values followed by the same letters do not differ significantly in Tukey's (HSD) test ($p < 0.05$). PA—*Paenibacillus azotofixans*; BM—*Bacillus megaterium*; BS—*Bacillus subtilis*.

According to Govindasamy et al. [28], a higher SPAD index value results from the provision of a sufficient amount of N, supported by microbial preparations, which enables the improvement of the condition of plants. An increase in nitrogen fertilisation is accompanied by an increase in both the leaf greenness index value and the leaf area index (LAI) [77]. Islam et al. [78] and Monostori et al. [79] also noted that nitrogen fertilisation significantly contributes to an increase in the flag leaf SPAD index value, which is positively correlated with the wheat grain yield. In the present study, no interaction was noted between the years of study and the preparations applied to the SPAD index values (data not shown).

The photosynthetic process rate positively correlates with the mineral content in the soil [80,81]. The photosynthetic intensity, however, can be impaired by almost any adverse environmental factor [81,82]. One such factor is a soil water deficit, which results from the weather condition pattern. Abiotic stresses primarily reduce the photosynthetic performance of plants due to their adverse effect on chlorophyll biosynthesis, photosystem performance, electron transport mechanisms, gas exchange parameters and many more [57].

On both measurement dates, the years of the study unevenly affected the photosynthetic rate shaping (Figure 4), and on both observation dates, irrespective of the years, the highest photosynthetic rate was noted following the application of NPK + *P. azotofixans* and NPK and all the preparations combined (*P. azotofixans* + *B. megaterium* + *B. subtilis*). The weather conditions throughout the years of the study differentiated the effect of biological preparations on the photosynthesis index. (Table 5). Such an example, as compared to the control plot (NPK), is obtaining lower photosynthetic rate values at the $BBCH_{31}$ stage following the application of NPK + *B. megaterium* in 2018 and at the $BBCH_{51}$ stage in 2018 and 2019. This was confirmed by the strong positive correlation between the grain yield and the net photosynthetic intensity, as noted in a study by Olszewski et al. [82], Carmo-Silva et al. [83], Murchie et al. [84] and Sanchez-Bragado et al. [85]. Such a relationship is indicative of the correct use of photosynthetic products in yield formation [86]. In

the present study, the regression equations describing the relationship of the photosynthetic rate at the wheat development stages of $BBCH_{31}$ and $BBCH_{51}$ are characterised by a determination coefficient of 0.40–0.99 (Table 6).

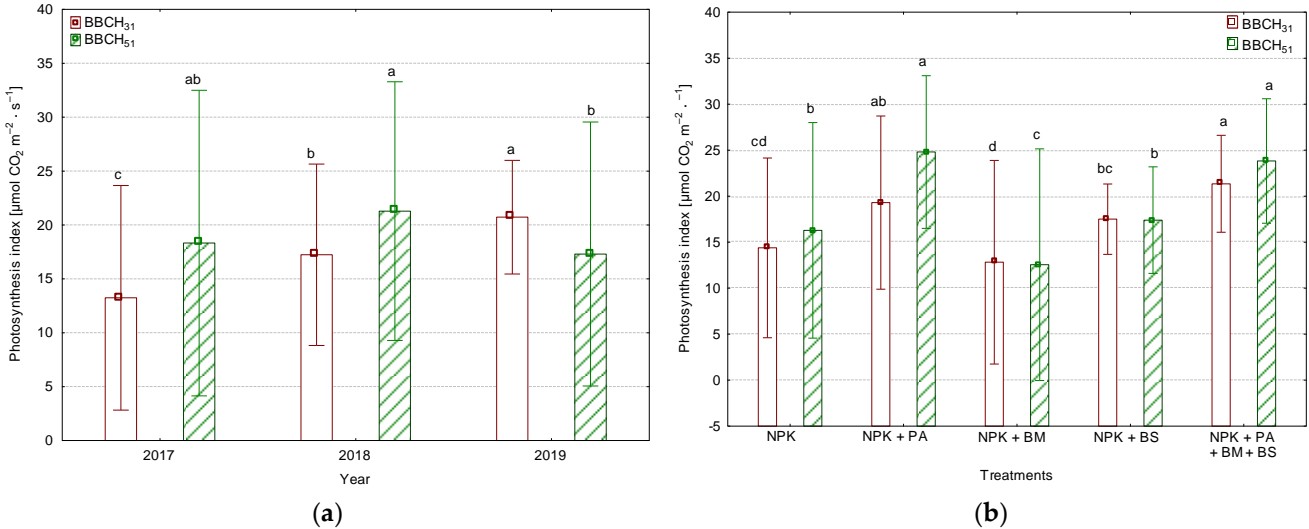

**(a)**                                                   **(b)**

**Figure 4.** The photosynthetic index after application of microbial preparations (mean value and standard deviation), (**a**) average for the years, (**b**) average for treatment. Values followed by the same letters do not differ significantly in Tukey's (HSD) test ($p < 0.05$). PA—*Paenibacillus azotofixans*; BM—*Bacillus megaterium*; BS—*Bacillus subtilis*.

**Table 5.** The photosynthetic index after application of microbial preparations (mean value and standard deviation), interaction between the years and treatments ($\mu mol\ CO_2\ m^{-2}\ s^{-1}$).

| Year | NPK | NPK + PA * | NPK + BM ** | NPK + BS *** | NPK + PA + BM + BS |
|---|---|---|---|---|---|
| | | | $BBCH_{31}$ | | |
| 2017 | 8.18 ± 3.04 [e] | 13.14 ± 0.46 [cde] | 9.33 ± 5.35 [e] | 15.58 ± 1.14 [bcd] | 20.02 ± 3.03 [ab] |
| 2018 | 17.23 ± 1.00 [bc] | 20.99 ± 0.29 [ab] | 10.57 ± 4.03 [de] | 17.17 ± 0.38 [bc] | 20.25 ± 2.11 [ab] |
| 2019 | 17.75 ± 0.07 [bc] | 23.77 ± 0.06 [a] | 18.58 ± 0.33 [abc] | 19.76 ± 0.16 [ab] | 23.77 ± 0.06 [a] |
| | | | $BBCH_{51}$ | | |
| 2017 | 10.22 ± 0.91 [de] | 27.07 ± 5.15 [a] | 16.85 ± 4.10 [bcd] | 13.75 ± 0.65 [cde] | 23.74 ± 3.81 [ab] |
| 2018 | 19.53 ± 7.13 [abc] | 25.80 ± 0.52 [a] | 14.93 ± 6.07 [cd] | 19.12 ± 1.92 [a–d] | 27.08 ± 0.21 [a] |
| 2019 | 19.12 ± 0.71 [a–d] | 21.55a ± 3.75 [bc] | 5.89 ± 0.46 [e] | 19.33 ± 0.12 [abc] | 20.67 ± 0.06 [abc] |

* PA—*Paenibacillus azotofixans*; ** BM—*Bacillus megaterium*; *** BS—*Bacillus subtilis*. Values followed by the same letters do not differ significantly in Tukey's (HSD) test ($p < 0.05$).

At the wheat growth phase of BBCH31, when fertilising wheat with NPK alone, NPK + *P. azotofixans*, NPK + *B. megaterium* and NPK + *B. subtilis*, an increase in the photosynthetic rate value was obtained with the maximum amount of rainfall and at the maximum temperature (Figure 5). As regards wheat fertilisation with NPK + *P. azotofixans* + *B. megaterium* + *B. subtilis*, an increase in the photosynthetic rate value occurred with an increase in the amount of rainfall, irrespective of the temperatures occurring during the period of measurements (May). At the $BBCH_{51}$ stage, the maximum increase in the photosynthetic rate was obtained at the minimum amount of rainfall and at the maximum temperature in the fertilisation variants of NPK, NPK + *B. subtilis*, NPK + *B. megaterium* and NPK + *P. azotofixans* + *B. megaterium* + *B. subtilis*. As regards the fertilisation with NPK + *P. azotofixans* at the BBCH51 stage, an increase in the photosynthetic rate was obtained with a minimum amount of rainfall and at a minimum temperature.

**Table 6.** Regression equations describing the photosynthetic rate (A) depending on the rainfall (R) and the temperature (T) at the winter wheat development stages (BBCH$_{31}$ and BBCH$_{51}$) after application of microbial preparations.

| Treatments | BBCH$_{31}$ | | BBCH$_{51}$ | |
| --- | --- | --- | --- | --- |
| | Regression Equation | Coefficient of Determination R$^2$ | Regression Equation | Coefficient of Determination R$^2$ |
| NPK | A = 0.0921·R + 2.5164·T − 24.5848 | 0.883 | A = −0.1753·R + 3.7715·T − 35.9334 | 0.585 |
| NPK + PA * | A = 0.0996·R + 2.1819·T − 15.7609 | 0.996 | A = −0.0160·R − 1.0631·T + 44.9450 | 0.354 |
| NPK + BM ** | A = 0.0801·R + 0.3463·T + 3.1292 | 0.599 | A = −0.0479·R − 1.9456·T + 50.9656 | 0.629 |
| NPK + BS *** | A = 0.0374·R + 0.4448·T + 9.2531 | 0.889 | A = −0.0972·R + 2.2360·T − 14.1208 | 0.867 |
| NPK + PA + BM + BS | A = 0.0321·R + 0.0641·T + 18.4380 | 0.464 | A = −0.1206·R + 0.5517·T + 24.0601 | 0.653 |

* PA—Paenibacillus azotofixans; ** BM—Bacillus megaterium; *** BS—Bacillus subtilis.

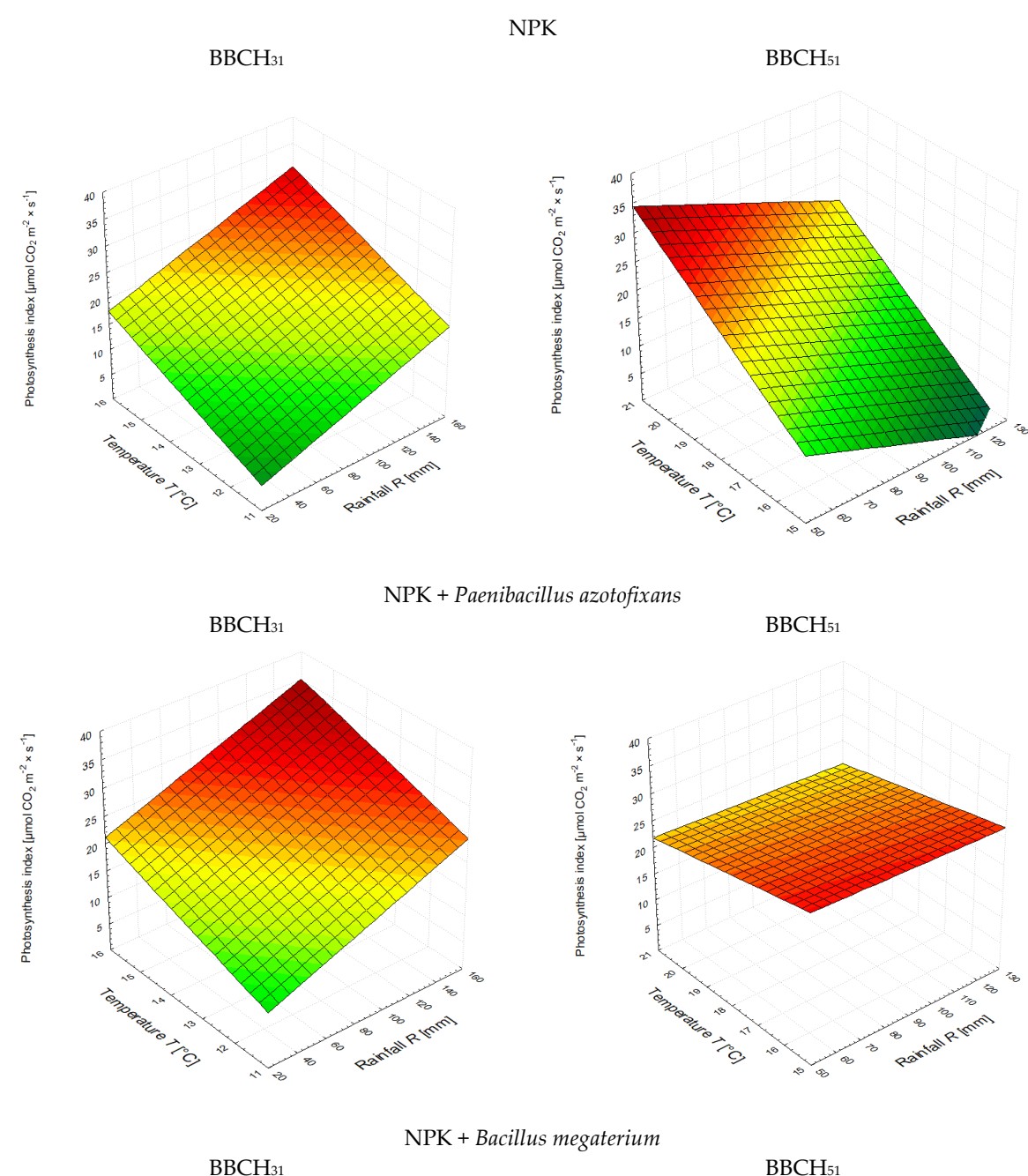

**Figure 5.** *Cont.*

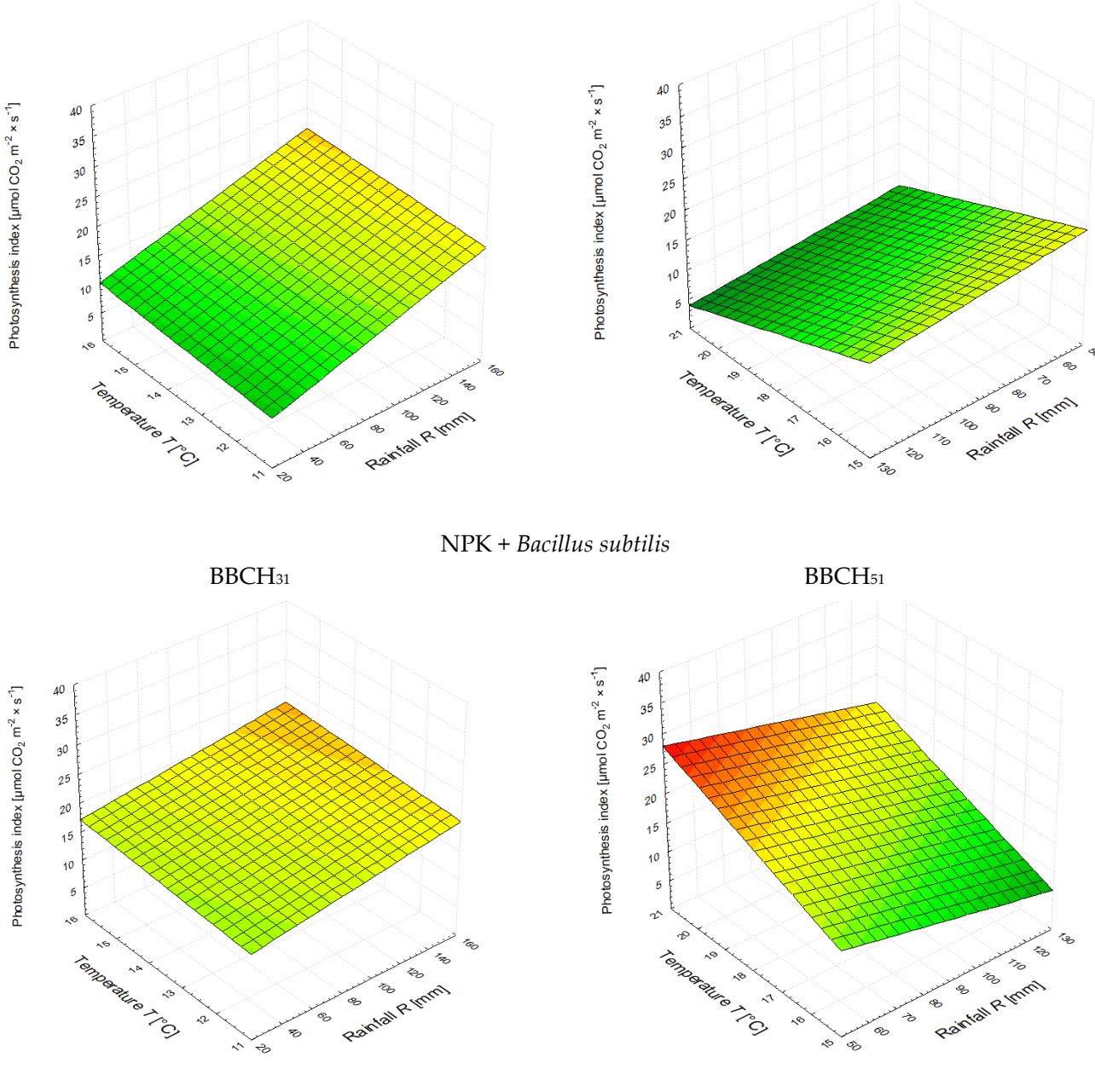

NPK + *Bacillus subtilis*

BBCH$_{31}$           BBCH$_{51}$

NPK + *Paenibacillus azotofixans* + *Bacillus megaterium* + *Bacillus subtilis*

BBCH$_{31}$           BBCH$_{51}$

**Figure 5.** *Cont.*

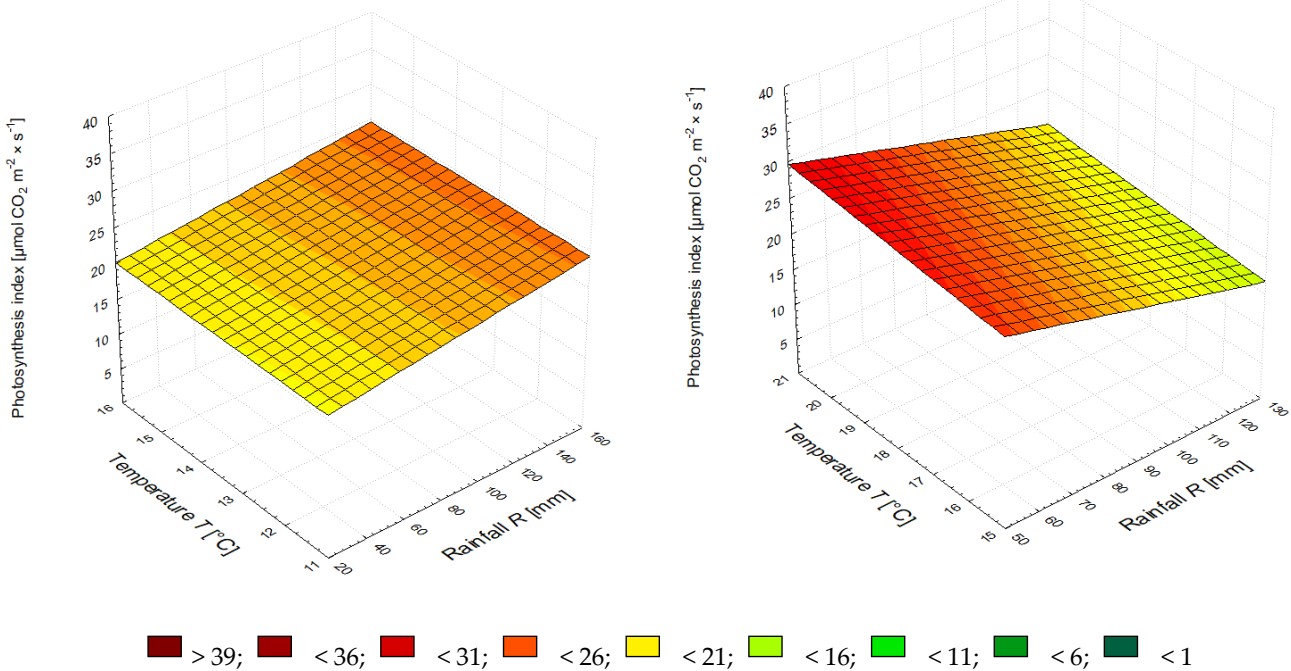

**Figure 5.** The photosynthetic index depending on the temperature and rainfall at the stages $BBCH_{31}$ and $BBCH_{51}$ after application of microbial preparations.

### 3.3. N-NO$_3$, N-NH$_4$, and P Contents in the Soil

The highest mineral nitrogen (N-NO$_3$ and N-NH$_4$) contents in the soil were noted in 2017 and were, on average, higher as compared to the subsequent years of the study by 43.3% and 45.5%, respectively (Figure 6). Irrespective of the years, an increase in the N-NH$_4$ content in relation to the control plot (NPK) was noted following the introduction of each plot. An increase in the N-NO$_3$ content in relation to the control plot was noted following the application of NPK + *P. azotofixans* and NPK and all the preparations combined (*P. azotofixans* + *B. megaterium* + *B. subtilis*). In a study by Kołodziejczyk [87], microbial preparations Proplantan AM and Effective Microorganisms significantly reduced the N-NO$_3$ and N-NH$_4$ contents in the soil. In the authors' own study, the principle of N-NO$_3$ content following the application of NPK + *P. azotofixans* being higher than that on the control plot (NPK) was not reflected in all the years of the study (Table 7), with an increase in the N-NH$_4$ content in relation to the control plot (NPK) noted in 2017 following the application of NPK + *P. azotofixans* and NPK + *B. subtilis*. The N-NH$_4$ contents were also significantly higher in 2018 and 2019 than those on the control plot (NPK) and were only found following the application of NPK and all of the microbial preparations in combination (*P. azotofixans* + *B. megaterium* + *B. subtilis*).

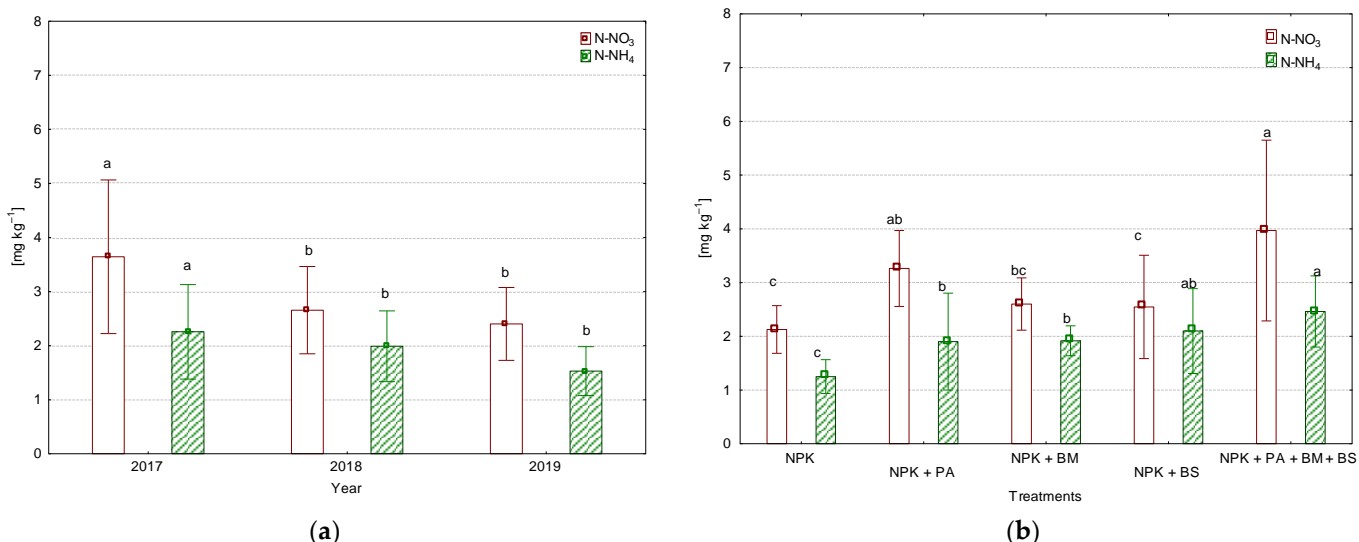

**Figure 6.** The N-NO$_3$ and N-NH$_4$ contents in the soil after application of microbial preparations (mean value and standard deviation), (**a**) average for the years, (**b**) average for treatment. Values followed by the same letters do not differ significantly in Tukey's (HSD) test ($p < 0.05$). PA—*Paenibacillus azotofixans*; BM—*Bacillus megaterium*; BS—*Bacillus subtilis*. [a,b,c]—statistically homogenous groups, $p \leq 0.05$.

**Table 7.** N-NO$_3$ and N-NH$_4$ contents in the soil after application of microbial preparations (mean value and standard deviation), interaction between the years and treatments.

| Year | NPK | NPK + PA * | NPK + BM ** | NPK + BS *** | NPK + PA + BM + BS |
|---|---|---|---|---|---|
| | | | **N-NO$_3$** | | |
| 2017 | 2.32 ± 0.52 [bc] | 3.79 ± 0.45 [b] | 2.77 ± 0.43 [bc] | 3.30 ± 0.57 [bc] | 6.05 ± 0.73 [a] |
| 2018 | 1.82 ± 0.48 [c] | 3.25 ± 0.81 [bc] | 2.82 ± 0.45 [bc] | 2.60 ± 1.17 [bc] | 2.80 ± 0.45 [bc] |
| 2019 | 2.25 ± 0.18 [bc] | 2.75 ± 0.51 [bc] | 2.22 ± 0.42 [c] | 1.75 ± 0.33 [c] | 3.05 ± 0.96 [bc] |
| | | | **N-NH$_4$** | | |
| 2017 | 1.12 ± 0.11 [d] | 2.97 ± 0.81 [a] | 2.04 ± 0.14 [a–d] | 3.03 ± 0.52 [a] | 2.13 ± 0.78 [a–d] |
| 2018 | 1.48b ± 0.50 [cd] | 1.46 ± 0.07 [bcd] | 2.15 ± 0.01 [abc] | 1.88 ± 0.02 [bcd] | 2.99 ± 0.58 [a] |
| 2019 | 1.15 ± 0.07 [cd] | 1.29 ± 0.21 [bcd] | 1.56 ± 0.08 [bcd] | 1.38 ± 0.33 [bcd] | 2.27 ± 0.32 [ab] |

* PA—*Paenibacillus azotofixans*; ** BM—*Bacillus megaterium*; *** BS—*Bacillus subtilis*. Values followed by the same letters do not differ significantly in Tukey's (HSD) test ($p < 0.05$).

In 2018, the P content in the soil was lower by an average of 16.8% than that noted in 2017 and 2019 (Figure 7). Irrespective of the year, the highest P content, similar to N-NO$_3$ and N-NH$_4$, was noted following the application of NPK and all the microbial preparations in combination (*P. azotofixans + B. megaterium + B. subtilis*). This effect of the applied NPK and all the biopreparations in combination was confirmed in all years of the study (Table 8). Furthermore, in 2018, a higher P content was noted after the application of NPK + *B. megaterium* than after the application of NPK alone. According to Kocoń and Jadczyszyn [88], favourable weather conditions (a high amount of evenly distributed rainfall and high temperatures) had a positive effect on the performance of microbial preparations (EM, EM-Farming, UGmax) on the available phosphorus content in the soil. *Bacillus* spp., which fixes atmospheric nitrogen and dissolves soil phosphorus resources, stimulates plant growth [23,37,89]. In a study by Turan et al. [60], preparations containing *Bacillus subtilis*, *Bacillus megaterium*, or *Azospirillum brasilense*, applied individually or in combination, increased the concentration of labile and moderately labile P fraction in the rhizosphere soil. Tahir et al. [90] found that mineral fertilisation in combination with a phosphorus-solubilising bacterial strain (*Bacillus strain* MWT-14) increased the P content by 3.3% and the N content by 3.7%, on average.

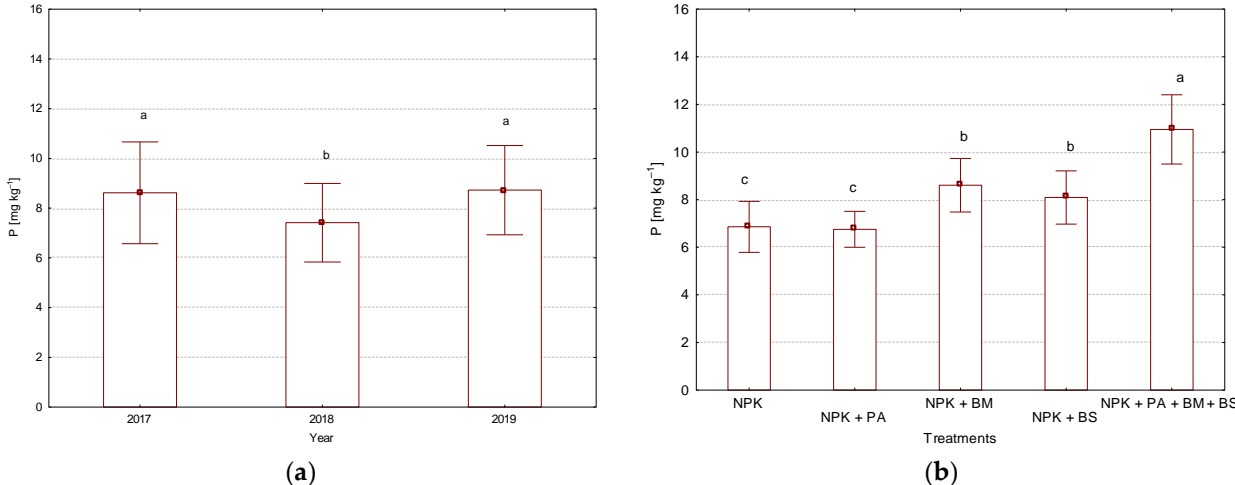

(**a**)  (**b**)

**Figure 7.** P content in the soil after application of microbial preparations (mean value and standard deviation), (**a**) average for the years, (**b**) average for treatment. Values followed by the same letters do not differ significantly in Tukey's (HSD) test ($p < 0.05$). PA—*Paenibacillus azotofixans*; BM—*Bacillus megaterium*; BS—*Bacillus subtilis*.

**Table 8.** P content in the soil after application of microbial preparations (mean value and standard deviation), interaction between the years and treatments.

| Year | NPK | NPK + PA * | NPK + BM ** | NPK + BS *** | NPK + PA + BM + BS |
|---|---|---|---|---|---|
| 2017 | 7.78 ± 1.34 [b–e] | 6.74 ± 0.84 [de] | 8.73 ± 1.24 [bcd] | 7.99 ± 1.52 [b–e] | 11.84 ± 0.24 [a] |
| 2018 | 5.91 ± 0.18 [e] | 6.14 ± 0.35 [e] | 8.54 ± 1.62 [bcd] | 7.26 ± 0.18 [b–e] | 9.24 ± 1.28 [b] |
| 2019 | 6.89 ± 0.09 [cde] | 7.39 ± 0.46 [b–e] | 8.56 ± 0.65 [bcd] | 9.03 ± 0.38 [bc] | 11.77 ± 0.48 [a] |

* PA—*Paenibacillus azotofixans*; ** BM—*Bacillus megaterium*; *** BS—*Bacillus subtilis*. Values followed by the same letters do not differ significantly in Tukey's (HSD) test ($p < 0.05$).

### 3.4. Principal Component Analysis–PCA

The analyses concerning the effect of the application of microbial preparations in winter wheat cultivation on the SPAD index, photosynthesis, grain yield, protein content in grains and the $N\text{-}NO_3$; $N\text{-}NH_4$; and P contents in the soil were supplemented by the determination of correlations between the above-mentioned factors (Figure 8). To this end, the principal component analysis (PCA) method was used to determine the links (the strength and direction of the correlation) between the measurement variables. The analysis showed a strong correlation between the leaf greenness index (SPAD) as well as photosynthesis and the protein content in wheat grains during both development stages ($BBCH_{31}$ and $BBCH_{51}$). The $N\text{-}NO_3$, $N\text{-}NH_4$ and P contents in the soil were also strongly correlated with the grain yield.

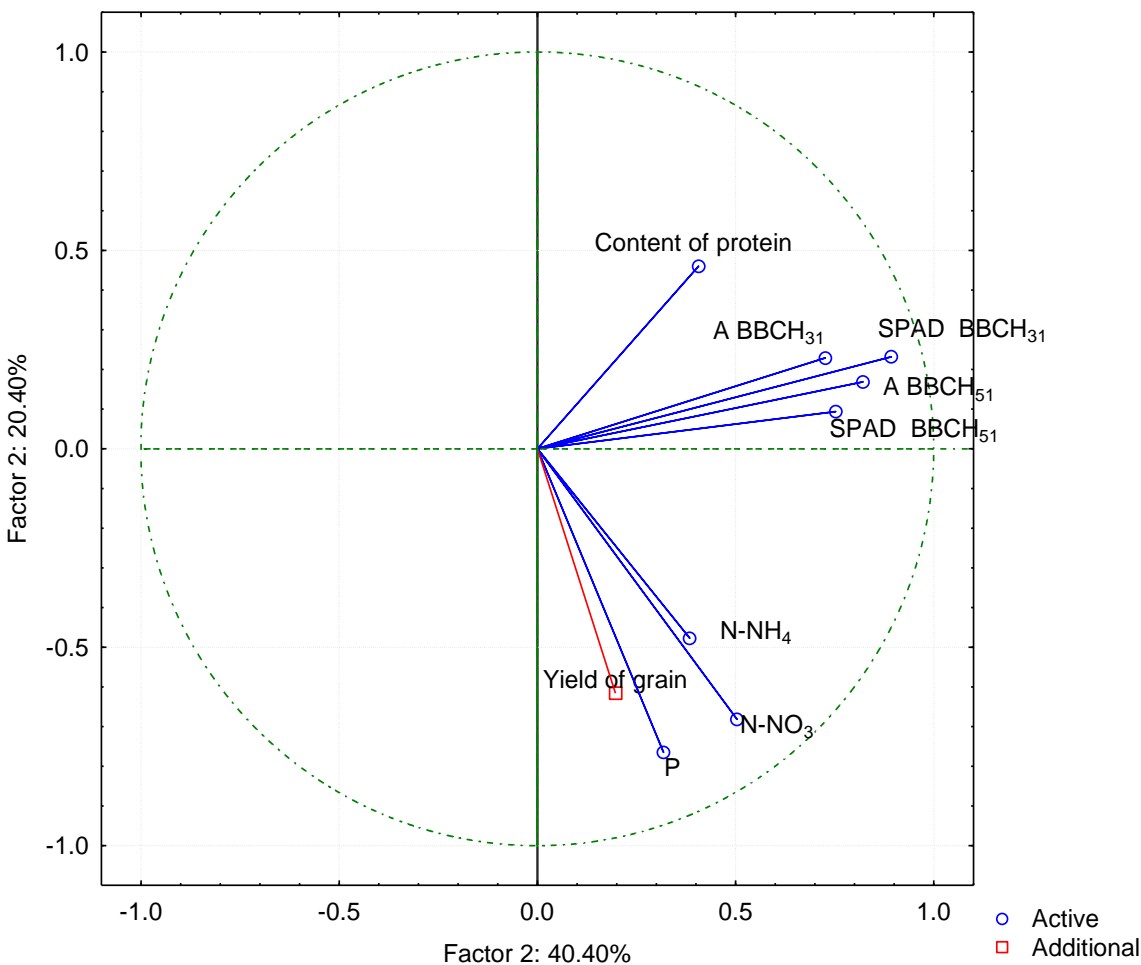

**Figure 8.** Diagram illustrating the correlation between the photosynthetic index (A), SPAD, protein content in grains, N-NO$_3$, N-NH$_4$ and P contents in the soil and the winter wheat grain yield.

## 4. Conclusions

The highest grain yield was noted following the application of mineral fertilisation and the three microbial preparations combined (*Paenibacillus azotofixans*, *Bacillus megaterium* and *Bacillus subtilis*), as well as NPK with *Paenibacillus azotofixans*. The preparation containing *Paenibacillus azotofixans* performed well under optimal conditions for wheat growth and development, while all the preparations applied in combination (*P. azotofixans*, *B. megaterium*, *B. subtilis*) performed well under extreme weather conditions by supplementing each other. The microbial preparations had a significant effect on the leaf greenness index (SPAD) at both test dates (BBCH$_{31}$ and BBCH$_{51}$). No interaction was noted between the years of study and the preparations applied on the SPAD values. The highest leaf photosynthetic index at both observation dates was noted following the application of NPK + *P. azotofixans* and NPK and all the preparations combined (*P. azotofixans*, *B. megaterium* and *B. subtilis*). At the development phase of BBCH$_{31}$, when fertilising wheat with NPK alone and applying NPK and the microbial preparations individually, an increase in the photosynthetic rate value was obtained at the maximum amount of rainfall and at the maximum temperature. At the BBCH$_{51}$ stage, the maximum increase in the photosynthetic rate was obtained with the minimum amount of rainfall and at the maximum temperature in the fertilisation variants of NPK, NPK + *B. subtilis*, NPK + *B. megaterium* and NPK + all the preparations in combination. The highest N-NO$_3$, N-NH$_4$ and P contents in the soil were noted following the application of NPK and all microbial preparations combined. Strong correlations were noted between the leaf greenness index (SPAD) and the photosynthetic index value and

the protein content in wheat grains and between the N-NO₃, N-NH₄ and P contents in the soil and the wheat grain yield.

**Author Contributions:** Conceptualisation, A.S., K.W. and E.K.; methodology, A.S., K.W. and E.K.; validation, A.S., K.W. and E.K.; formal analysis, A.S., K.W. and E.K.; investigation, A.S.; resources, A.S., K.W. and E.K.; data curation, A.S., K.W. and E.K.; writing—original draft preparation, A.S., K.W. and E.K.; writing—review and editing, A.S., K.W. and E.K.; visualisation, A.S. and E.K.; supervision, A.S., K.W. and E.K.; project administration, A.S.; funding acquisition, A.S., K.W. and E.K. All authors have read and agreed to the published version of the manuscript.

**Funding:** Project financially supported by the Minister of Education and Science under the program entitled "Regional Initiative of Excellence" for the years 2019–2023, Project No. 010/RID/2018/19, amount of funding 12.000.000 PLN. The results presented in this paper were obtained as part of a comprehensive study financed by the University of Warmia and Mazury in Olsztyn, Faculty of Agriculture and Forestry, Department of Agroecosystems and Horticulture (grant No. 30.610.015-110).

**Institutional Review Board Statement:** Not applicable.

**Informed Consent Statement:** Not applicable.

**Data Availability Statement:** Not applicable.

**Conflicts of Interest:** The authors declare no conflict of interest.

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
