# Peer review of "Effect of Commercial Microbial Preparations Containing Paenibacillus azotofixans, Bacillus megaterium and Bacillus subtilis on the Yield and Photosynthesis of Winter Wheat and the Nitrogen and Phosphorus Content in the Soil"

_applsci, doi:10.3390/app122412541_

Round 1

Reviewer 1 Report

The present study is carried out according to a well-established design, specific to a group of experienced specialists in the field of fertilizers.

The information from the literature and the punctual results are processed and depicted in a clear and concise form, which means that <practically there are no more questions>.

Author Response

We would like to thank the Honorable Reviewer for his time and valuable comments.

Reviewer 2 Report

This study has assessed the effects of commercially available microbial preparations/formulations on the grain yield and photosynthetic efficacy of winter wheat and the nutrient content of the soil as compared to recommended synthetic fertilizer applications. While this study has provided some novel insights into the effects of the application of such microbial biofertilizers for sustainable wheat production, the manuscript cannot be recommended for publication until revised. Thus, the manuscript needs improvement to meet the journal's standards. In particular, English editing is highly encouraged.

Author Response

We would like to thank the Honorable Reviewer for his time and valuable comments.

The text was corrected according to the reviewers' instructions.

The text was translated by the Translation Agency which issued the certificate. The native speaker re-examined the text and made the necessary corrections.

Reviewer 3 Report

Dear Editor, thank you for this opportunity.

The research article is well-written and the experiment looks promising. It is an interesting research work and could contribute to the field of biological control and integrated pest management.  Please find my comments on the paper in the attached document.

best regards

Author Response

The research article is well-written and the experiment looks promising. It is an interesting research work and could contribute to the field of biological control and integrated pest management.  Please find my comments on the paper in the attached document

We would like to thank the Honorable Reviewer for his time and valuable comments.

Response:

The text was corrected according to the reviewers' instructions.

The text was translated by the Translation Agency which issued the certificate. The native speaker re-examined the text and made the necessary corrections.

Line 17 – was?

Response:

The text was corrected according to the reviewers' instructions,  and corrections by a native speaker

Line 26 - Please avoid repetitive expressions.

Response:

The text was corrected according to the reviewers' instructions.

Line 51, 54, 56, 74, 78, 91, 208, 209, 214, 224 -italic, please check throughout the manuscript

Response:

The text was corrected according to the reviewers' instructions.

Line 51  - I also wonder Did you check the presence of these bacteria in field study areas?

Line 51 - Please add related references

Response:

Line 51. This was not part of our research, and neither are our statements. At the suggestion of another reviewer, we removed the fragment: „They are isolated from the soil, water, the digestive tract of animals, and food”

Line 60 - Bacillus subtilis is a saprophyte bacterium which.

Response:

We remove: „The bacterium:

Line 90 - Please add a transition sentence before the last paragraph

Response:

A transition sentence has been inserted:

"The use of biological preparations in the form of biofertilisers may be a way to ensure environmental stability in intensive agricultural production."

Line 98 - The experiments were carried out in the years 2017-2019 using commercial bacterial preparations that are currently used in the cultivation of winter wheat (Triticum aestivum L.).

Response:

Line 98 - As suggested by the Reviewer, corrected to:

“ The experiments were carried out in the years 2017-2019 using commercial bacterial preparations that are currently used in the cultivation of winter wheat (Triticum aestivum L.).”

Line 105 - Wheat? please state clearly

Response:

The text has been corrected according to the reviewer's instructions.

Inserted word "wheat"

Line 110 - Weather conditions registered during field trials are presented in Table 1.

Response:

The text has been corrected according to the reviewer's instructions on:

“Weather conditions registered during field trials are presented in Table 1.”

Line 112 - You can also move this table to Results section

Response:

Line 112 - Thank you for your attention. However, we believe it is not necessary to move the table to the Results section. We would have to add a new subsection, which would destroy the concept of the work and we would be forced to re-edit the work, and possibly new reviews.

Line 113 (Table 1) - How did you measure the rainfall? A weather station? Please add a few sentences.

Response:

Lines 113

Information on the origin of meteorological data "data according to the Meteorogical Station at Tomaszkowo, (53o71'N, 20o43'E, Poland" has been supplemented in the table title

Reviewer 4 Report

In this manuscript, the author studied the affection of three bacterial fertilizers, Paenibacillus azotofixans, Bacillus megaterium, and Bacillus subtilis, on the growth of winter wheat. The application of the three bacteria promotes crop growth, including photosynthesis, yield, N and P content. However, there are still some problems the author should be noticed. Thus, the manuscript is suggested to be reconsidered after a major revision.

The problems including:

1. It was noticed that the total performance of NPK+PA was close to the NPK+PA+BM+BS, sometimes the indexes of NPK+PA were even higher than NPK+PA+BM+BS, indicating that using PA alone would achieve a good yield. Then why should we study the combining application of three bacteria? And what the advance of using three bacteria simultaneously?

2. It was known that the bacteria can affect the Fe content in crop and soil, it is recommended to show these data, which may be crucial to the evaluation of the bacterial fertilizers.

3. Most of the name of bacteria need to use italic font.

4. In Table 6 and Figure 5, the model used in the fitting may not be a linear function, other functions are recommended to be tried to improve the R2 value.

5. In Figure 4a, the statistic mark of the data of 2019 may be mistaken, please check. Also, the quality of all of the figures needs to be improved.

Author Response

In this manuscript, the author studied the affection of three bacterial fertilizers, Paenibacillus azotofixans, Bacillus megaterium, and Bacillus subtilis, on the growth of winter wheat. The application of the three bacteria promotes crop growth, including photosynthesis, yield, N and P content.

However, there are still some problems the author should be noticed. Thus, the manuscript is suggested to be reconsidered after a major revision. The problems including:

Response:

We would like to thank the Honorable Reviewer for his time and valuable comments.

In the work, we studied the effect of three bacterial fertilizers, Paenibacillus azotofixans, Bacillus megaterium and Bacillus subtilis. The use of these three bacteria supports the growth of crops, including photosynthesis, yield, N and P content. The strains of these bacteria are characterized by different properties, primarily with regard to the optimal conditions for their activity. Despite the close relationship of these microorganisms, they differ in their properties and activity. They can work differently when used individually than when used together. The practical recommendations for the use of these preparations include the possibility of their combined use. Therefore, our experiment was to verify the effects of the preparations separately and the need for their combined use.

  1. It was known that the bacteria can affect the Fe content in crop and soil, it is recommended to show these data, which may be crucial to the evaluation of the bacterial fertilizers.

Response:

In our work, we focused on the study of the impact of biological preparations on the availability of N and P necessary for plant growth and the most yield-producing. The bacteria contained in the tested biological preparations have a comprehensive effect on the improvement of the physical and chemical properties of the soil and on the growth of plants.

When planning the experiment, we took into account the information provided by the manufacturer who suggested the properties of Bacillus azotofixans.i Bacillus megaterium in improving the circulation of nitrogen and phosphorus in the soil. Therefore, in the work we referred only to the presentation of the elements listed in the title of the work.

  1. Most of the name of bacteria need to use italic font.

Response:

The text was corrected according to the reviewers' instructions.

  1. In Table 6 and Figure 5, the model used in the fitting may not be a linear function, other functions are recommended to be tried to improve the R2 value.

Response:

Performing a multiple regression analysis of the photosynthesis index depending on temperature and precipitation on the 31st and 51st day of cultivation, after checking the possibility of mapping the dependent variable as a function of independent variables, it was decided to use the simplest linear function model for two variables:

Z= Ax+By+C.

By checking other functions describing the photosynthesis index depending on the temperature and precipitation on the 31st and 51st day (also using forward and backward stepwise regression methods) in order to improve the value of the coefficient of determination R2, the value of the coefficient of determination was obtained the same as in the case of the obtained linear function or lower.

  1. In Figure 4a, the statistic mark of the data of 2019 may be mistaken, please check. Also, the quality of all of the figures needs to be improved.

Response:

Thank you for your attention, however, it seems to us that in Figure 4a for 2019 the statistics are presented correctly.

The quality of some of figures has been improved, however, if the Editor wishes, we can further increase the quality of the figures by increasing the size, but then 2 drawings would not fit next to each other.

Reviewer 5 Report

this work is very valuable and adds a significant contribution to the field of  agricultural biotechnology 

Author Response

We would like to thank the Honorable Reviewer for his time and valuable comments.

The text was corrected according to the other reviewers' instructions.

The text was translated by the Translation Agency which issued the certificate. The native speaker re-examined the text and made the necessary corrections.

Reviewer 6 Report

I would suggest rephrasing the title of this work may be: The Effects of Commercial Microbial Preparations Containing Paenibacillus azotofixans, Bacillus megaterium, and Bacillus subtilis on the Yielding and Photosynthesis of Winter Wheat and the Soil minerals 

The introduction is a bit lengthy and doesn't flow well. I would suggest restructuring it. 

Few editing issues: italics of scientific names are inconsistent, few commas are missing in the lists of items

Author Response

We would like to thank the Honorable Reviewer for his time and valuable comments

I would suggest rephrasing the title of this work may be: The Effects of Commercial Microbial Preparations Containing Paenibacillus azotofixans, Bacillus megaterium, and Bacillus subtilis on the Yielding and Photosynthesis of Winter Wheat and the Soil minerals

Response:

Regarding the proposed title change:

Thank you for this comment, however, we believe that the current title is adequate to the content of the work. The introduction of the phrase "Soil minerals" instead of "Nitrogen and Phosphorus Content in the Soil" is too general. The proposed title would imply that we intend to present results with multiple minerals, but in our paper we only show results for nitrogen and phosphorus.

The introduction is a bit lengthy and doesn't flow well. I would suggest restructuring it.

Response:

As suggested by the reviewer, we removed redundant fragments of the text in the Introduction.

Line 41 -  A number of commercial biological preparations are available on the market, including plant biostimulants (humic acids, protein hydrolysates, and seaweed extracts) and microorganisms (mycorrhizal fungi and rhizobacteria).

Line 71 -  Since this bacterium multiplies rapidly, it eliminates other pathogenic organisms naturally from the soil environment by competing with them for food and space without disturbing (and actually supporting) the development of beneficial soil microorganisms

Line -  Besides the other above-mentioned characteristics, the species belonging to the genus Paenibacillus have a significant characteristic of nitrogen fixation, which is not exhibited by most PGPR strains belonging to other genera.

Few editing issues: italics of scientific names are inconsistent, few commas are missing in the lists of items

Response:

The text was corrected according to the reviewer instructions.

Line 74,76,78, 205, 206, 208, 209, 213, 214, 216, 224, 251, 291, 296, 312, 313, 358, -italic

Response:

The text was corrected according to the reviewer instructions.

Line 102, 107 - upper case,

Response:

The text was corrected according to the reviewer instructions.

Line 299, 364  - add comma before 'and'

Response:

The text was corrected according to the reviewer instructions.

Round 2

Reviewer 2 Report

The manuscript provides some useful information about microbial bioformulation for winter wheat. But the way that manuscript is drafted makes it difficult to understand and reduces the quality of the work undertaken. I strongly suggest improving the manuscript by carefully addressing the comments given and also with English editing. 

Author Response

Thank you for reviewing our manuscript. We corrected the manuscript according to the comments of the Reviewer. We made the corrections in the Word system using "Track Changes". As this is the second round, additionally, to mark the new changes, we have marked the inserts in red.

Line 75 - Are these references for the previous sentence?

Answer:

We agree with the Reviewer. We moved position "33" to the right place, it is now position "35"

Line 75 = increase or involve?

Answer:

We corrected on involvecrease

Line 76 - What do you mean by soil growth

Answer:

We agree with the Reviewer. Incorrect term has been changed.

Line 91 - State the gap in the knowledge that led you to this study or to the aim that follows in the next paragraph.

Answer:

We agree with the Reviewer and we added a statement: “At the moment, there are no studies focused on the comprehensive effect of biological preparations on the photosynthesis processes, the yield of wheat grain, and the content of nitrogen and phosphorus in soil”

Line 114 - Please combine this sentence with the previous paragraph.

Answer:

We agree with the Reviewer and we combined it with the previous paragraph

Line 118 -  Data obtained from??

Answer:

We corrected for the version suggested by the Reviewer

Line 124 -  What is this sentence for? I hope it has lost its place. Please move it to an appropriate place rather than keeping a single sentence as a paragraph.

Answer:

The position of this statement has been changed, moved higher, as suggested by the Reviewer.

Line 135 -  Please follow my comments in my previous review report. Table titles should be self explanatory. It should give clear information of what is given in the table without referring to the text for meaning.

Answer:

In the previous review, the review attachment was not insert to the MDPI system and we could not comment on it. However, we have corrected the titles of the tables and graphs, identifying them with the results of the research.

Line 175 – by

Answer:

We corrected for the version suggested by the Reviewer

Line 180 - ????? R ?

Answer:

We corrected for the version suggested by the Reviewer

Line 180 -  analysed?

Answer:

We corrected for the version suggested by the Reviewer

Line 184 – ??? a

Answer:

We corrected for the version suggested by the Reviewer

Line 190 - Please rephrase this sentence.

Answer:

The correlation of the effect of the applied biological preparations on the leaf greenness index, photosynthesis, grain yield, protein content in grains, and the N-NO3, N-NH4 and P contents in the soil in wheat cultivation was also determined

Line 195 -  Should not you report the results in past tense?

Answer:

We entrusted the translation to a native speaker and we received an appropriate certificate of translation correctness from him.

Line 198 - Is this the reason for the 2008 yield reduction?

Answer:

We have added a relevant statement: “which could have resulted in lower yields in 2018”

Line 198 - the meaning is not clear. What you you mean by low water resources?

Answer:

We changed on: rainfall

Line 239 -  The meaning is not clear.

Answer:

All figures and tables were corrected as suggested by the Reviewer

Line 241 - You should mention the meaning of the bars not on the figure itself, but just after the figure title

Answer:

All figures and tables were corrected as suggested by the Reviewer

Line 242 - Not a good title for this table.

Answer:

All figures and tables were corrected as suggested by the Reviewer

Line 243  - Please follow my comments given for the previous figure.

Answer:

All figures and tables were corrected as suggested by the Reviewer

Line 251 -  It is always better to give your results or key finding and then start discussion. But, unfortunately, that flow is missing throughout this section.

Answer:

Thank you for your suggestions. In scientific publications, we have also encountered such a system of discussions, which we present in this publication.

In this publication, we have combined the results and discussion into one chapter. At the beginning we present a short introduction, then our own results and discussion.

Line 262 - Why? Discuss about this trend.

Answer:

We have corrected this fragment of the text to: The influence of microbial preparations on the protein content in wheat grain was not demonstrated.

Line 271 - Please follow my previous comment.

Answer:

All figures and tables were corrected as suggested by the Reviewer

Line 276 - Not a good title. Effects of something on something......

Answer:

Thank you for your suggestions. In scientific publications, we have also encountered such an arrangement of titles in subsections. While we agree that table titles should be informative without the need to read the text, subsection titles should be concise, and other information is available in the text.

Line 277 - Not clear.

Answer:

We corrected for the version suggested by the Reviewer. An excerpt has been added to the sentence: “Weather conditions during the research”

Line 290 - In the present study? what do you meany by own study? Also combine this with the previous paragraph

Answer:

We corrected as suggested by the Reviewer in “In the present study”. Also combine this with the previous paragraph.

Line 298 - Please follow my previous comments on figures to improve the title of this figure and others in the MS.

Answer:

All figures and tables were corrected as suggested by the Reviewer

Line 310-311 - Please rephrase.

Answer:

We changed this sentence to: “The weather conditions throughout the years of the study differentiated the effect of biological preparations on the photosynthesis index”

Line 318 -  please see my previous comment.

Answer:

We corrected as suggested by the Reviewer in “In the present study”.

Line 345-  The axis labels are not clear.

Answer:

We improved drawing quality. We moved the legend to one place which allowed the axis labels to be clearer.